# Metabolic Stress in the Transition Period of Dairy Cows: Focusing on the Prepartum Period

**DOI:** 10.3390/ani10081419

**Published:** 2020-08-14

**Authors:** Osvaldo Bogado Pascottini, Jo L. M. R. Leroy, Geert Opsomer

**Affiliations:** 1Department of Reproduction, Obstetrics and Herd Health, Faculty of Veterinary Medicine, Ghent University, 9820 Merelbeke, Belgium; geert.opsomer@ugent.be; 2Veterinary Physiology and Biochemistry, Department of Veterinary Sciences, University of Antwerp, 2610 Wilrijk, Belgium; jo.leroy@uantwerpen.be

**Keywords:** dry period, metabolic status, energy balance, systemic inflammation, insulin resistance, transition disease

## Abstract

**Simple Summary:**

Complex pathways of metabolic adaptation occur in high-yielding dairy cows around calving. These adaptations require the redirection of nutrients to support the last stages of fetal growth and the commencement of lactation. Failure to adapt to these changes may result in the development of clinical disease in the postpartum period. Therefore, most existing literature is focused on studying the metabolic changes in the postpartum period. However, some of the risk factors associated with postpartum clinical disease can already be found in the prepartum period. This review describes adaptive changes occurring in prepartum high-yielding dairy cows, from drying off (40 to 60 days before parturition) until calving.

**Abstract:**

All modern, high-yielding dairy cows experience a certain degree of reduced insulin sensitivity, negative energy balance, and systemic inflammation during the transition period. Maladaptation to these changes may result in excessive fat mobilization, dysregulation of inflammation, immunosuppression, and, ultimately, metabolic or infectious disease in the postpartum period. Up to half of the clinical diseases in the lifespan of high-yielding dairy cows occur within 3 weeks of calving. Thus, the vast majority of prospective studies on transition dairy cows are focused on the postpartum period. However, predisposition to clinical disease and key (patho)physiological events such as a spontaneous reduction in feed intake, insulin resistance, fat mobilization, and systemic inflammation already occur in the prepartum period. This review focuses on metabolic, adaptive events occurring from drying off until calving in high-yielding cows and discusses determinants that may trigger (mal)adaptation to these events in the late prepartum period.

## 1. Introduction

The dairy cow transition period consists of a complex interplay of multiple pathways, including metabolic and hormonal adaptations, inflammation, and immune activation. These changes occur in a chain reaction fashion that begins within three weeks before calving and lasts for three to four weeks after parturition [1,2,3]. However, most prospective studies on transition dairy cows are focused on events occurring in the postpartum period [4]. This is because the most radical physiological changes in the transition period—such as calving, uterine involution, and the commencement (and maintenance) of lactation—happen in the postpartum period. Thus, the vast majority of metabolic and infectious diseases of dairy cattle occurs during this period [5]. However, some of the risk factors associated with postpartum clinical disease already originate before calving. In this regard, studies involving prepartum cows often evaluate risk factors for transition diseases focused on determining associations between risk factors (e.g., metabolic markers) and disease outcomes in the postpartum period. This approach typically entails selecting variables based on their potential causal association with the outcome [6]. However, they lack substance in describing the physiological changes that trigger variations in metabolic markers in the prepartum period. This review explores key (patho)physiological events occurring from drying off until calving in high-yielding (mostly) Holstein cows and discusses determinants that may trigger changes in the metabolic and inflammatory profile in the prepartum period. For the reader’s convenience, this paper is structured in four sections: (1) dry-off, (2) dry period feeding strategies and length, (3) a spontaneous reduction in feed intake, and (4) the body condition score at the end of pregnancy.

## 2. Drying Off: It All Starts Here

The dry period is necessary to allow udder tissue regeneration in order to guarantee optimal milk production in the next lactation [7,8]. A 40- to 60-day-long dry period has been the gold standard of management for dairy cows for the last 30 years [9]. The management of the dry period and factors affecting mammary gland biology and their implications (including for dairy cow welfare) have been thoroughly reviewed [10,11]. Only a summary and selective discussion of local and systemic changes associated with the dry-off will be added here.

Nutritional research has led to the formulation of diets that promote high milk production [12]. These diets, combined with improvements in animal genetics, have exponentiated the milk yield of modern dairy cows, making the cessation of lactation more challenging [10]. The dry-off begins approximately one week before the abrupt cessation of milking, by restricting energy intake. Rations with low energy content can be achieved by reducing concentrate levels, increasing less-palatable forage, or feeding (only) forage [11]. This protocol is somewhat standard in most modern dairy farms. The main challenge is to achieve the end goal (reduced milk production) without compromising the cow’s metabolic health [13]. One of the main factors associated with the dry-off success is the amount of milk the cows are producing at the desired moment of milk cessation. The aim is to decrease the daily milk yield to less than 12.5 kg and to subsequently stop milking abruptly [14]. However, when milking is ceased abruptly, the cisternal ducts and alveoli of the udder become engorged, raising the intramammary pressure (Figure 1) [15]. These events induce the mammary gland involution process [16], and when milk is no longer removed from the gland, prolactin production ceases, triggering apoptosis (Figure 1) [16,17]. Continuous epithelial cell proliferation and/or apoptosis are controlled by both systemic and local factors. Serotonin (5-hydroxytryptamine), a neurotransmitter synthesized from tryptophan, has been described as a feedback inhibitor of lactation. However, there is a general lack of research on factors associated with the local regulation of bovine mammary gland apoptosis and involution. It is widely accepted that mammary glands with lesser engorgement do exhibit an inflammatory response; however, the types and numbers of cells involved in this inflammatory response might be different when the milk yield at the dry-off is high [18].

Putman et al. [20] described biomarkers associated with metabolic stress and inflammation in blood samples collected at multiple days relative to dry-off. They found greater serum non-esterified fatty acids (NEFA) and cortisol concentrations within 2 days of drying off. The lipolysis may be associated with energy restriction (the cows switched to a dry cow diet after drying off) and/or fasting due to regrouping cows into their new pen (and cortisol due to social stress) the day after drying off. The high serum NEFA concentrations on the day after drying off may also be associated with an inflammatory response (and pain) due to increased intramammary pressure. In this regard, inflammation has been described as exacerbating lipid mobilization [21,22]. Changes associated with serum β-hydroxy butyrate (BHB) concentrations were less marked than those for NEFA. Still, high serum BHB concentrations were also found within 2 days of drying off. Interestingly, haptoglobin (positive acute phase protein) only peaked 12 days after drying off, indicating a systemic inflammatory response, probably associated with the mammary gland involution process. In general, the authors concluded that after drying off, there were systemic changes in metabolic (NEFA and BHB) and inflammatory (haptoglobin) markers. The metabolic peak occurred within 2 days and the inflammatory peak within 2 weeks of the abrupt cessation of milking. However, these changes were not equivalent in magnitude nor duration to those seen during the periparturient period. In a recent study, Mezzetti et al. [19] dug further and described several inflammatory and behavioral changes in cows producing ≥15 vs. <15 kg milk daily at the dry-off. They showed that both groups had increased plasma concentrations of liver enzymes, serum amyloid A and ceruloplasmin, and nitrogen species (nitrite, nitrate, and nitric oxide) within the week after milk cessation. Notably, the rise in these metabolic markers was exacerbated in cows producing ≥15 kg milk at the day of drying off. Furthermore, cows that produced ≥15 kg milk per day had a greater dry matter intake (DMI) and rumination time than those producing <15 kg, but both groups decreased their feed intake after the dry-off.

It is important to highlight that studies focusing on metabolic changes at the dry-off are scarce, and although the two studies mentioned above are novel, they have relatively small sample sizes. It is clear that the level of milk yield at drying off and the dry-off itself are important factors associated with metabolic and inflammatory changes in the prepartum period. However, it remains unknown whether the magnitude of these changes merely represents a physiological response, whether they negatively affect the cows’ health, or whether there is a threshold at which these biomarkers would predict disease. Additionally, it would be interesting to evaluate further the effects of co-variables such as body condition score (BCS), lactation number, and somatic cell counts in the previous lactation on metabolic and inflammatory changes at the dry-off. Future studies should consider these co-variables and focus on the intensity and consequences of metabolic and inflammatory changes in further stages of the dry and the postpartum period.

## 3. Feeding Strategies and Dry Period Length: Effects on Metabolism and Systemic Inflammation

Due to the abundant existing literature reviewing feeding management strategies and dry period length [23,24,25,26], only some general aspects will be discussed in this section.

### 3.1. Feeding Controlled-Energy and Negative Dietary Cation–Anion Difference Diets in the Prepartum Period

The feeding management of cows with a dry period of 40 to 60 days involves two diets: far off and close up. Changes in the diet and grouping of cows may contribute to metabolic changes associated with the dry-off. The far-off low-energy-density diet is designed to maintain the cow’s body condition and is generally delivered during the first 5 weeks of the dry period [27]. Subsequently, the close-up diet, with moderate energy density, is delivered during the final 3 weeks of the dry period [24]. This is designed to optimize the adaptation of the ruminal microorganisms to the high-energy diets provided just after parturition. Feeding a single diet throughout the dry period may contribute to reducing the metabolic stress and its impact on physiological parameters during the dry period and subsequent lactation. Although feeding a low-energy diet during the whole dry period can be successful, the potential impact of a sudden shift from low- to high-energy diets on postpartum cows and the concomitant adaptation of the ruminal microflora is a major challenge [24].

A high energy diet fed in the far-off period improves prepartum DMI, but it was also shown to increase prepartum BCS with its concomitant adverse effects on postpartum metabolic stress [28,29]. Current research focuses on controlling the energy intake in the dry period (1.30 to 1.39 Mcal of NEL/kg of DM) [26]. Feeding low energy diets in the prepartum period has repeatedly been associated with higher DMI and enhanced metabolic status in early lactation [28,29,30]. Controlling the prepartum energy intake can be achieved by feed restriction or by incorporating low energy ingredients in the diet in an ad libitum feeding fashion. The second strategy is preferred since feed restriction may lead to increased competition at the feed bunk in free-stall management systems [30] and because the rumen function may also benefit from the addition of additional fiber to the diet. The metabolic benefit of feeding a controlled energy diet in the prepartum period has been associated with lower blood NEFA, BHB, and liver triacylglycerols in the early postpartum period compared to those in cows fed a high-energy diet [30,31]. Nevertheless, the incorporation of controlled energy diets in the prepartum may introduce a sorting behavior in favor of smaller particles (with high energy density) that may lead to cows consuming a diet with higher energy than intended. Besides, the sorting behavior may persist in the fresh period with eventual negative consequences. Havekes et al. [32,33,34] very recently proposed a variety of strategies to reduce the sorting behavior in low energy diets in the dry period. In a first experiment [32], they demonstrated that prepartum cows being fed a short chop length (2.54 cm screen) of wheat straw is preferable since it was associated with lower serum BHB concentration in the third week postpartum compared to that in cows fed a long chop length (10.16 cm screen) of wheat straw. Although a shorter chop length of straw reduced sorting, it did not (completely) eliminate this behavior. In a next set of experiments, Havekes et al. [33,34] moisturized the content of high-straw dry cow diets with water or liquid molasses. The hypothesis was that the addition of water to dry-cow diets that are low in moisture would improve DMI and reduce feed sorting, while the addition of liquid molasses will additionally support ruminal cellulolytic microbial growth. Although moisturizing high-straw prepartum diets with either water or liquid molasses increased the prepartum DMI, this was not translated into an improved metabolic status (NEFA, BHB, and haptoglobin) in the postpartum period.

An up-to-date review of the literature on calcium (Ca) metabolism and negative dietary cation–anion difference (DCAD) diets can be found in the Ph.D. thesis of Couto Serrenho [35] and in other recent publications [26,36,37]. Briefly, the blood Ca concentration starts decreasing in the 1 to 2 days before calving [38,39] and reaches its nadir within 48 h of parturition [40,41,42]. The prevalence of clinical hypocalcemia has decreased in the last few years to a lactational incidence risk of <2.5% [43]. However, subclinical hypocalcemia (the cut-points vary from blood Ca <1.95 to <2.14 mmol/L, in the 24 to 48 h after calving) remains a significant problem affecting over half of multiparous Holstein cows [35]. Subclinical hypocalcemia is mostly associated with impaired postpartum health and (probably) with decreased milk yield [36,37]. A valid strategy to reduce the incidence of subclinical hypocalcemia is feeding prepartum DCAD diets (˃−100 mEq/Kg) [36,37]. Negative DCAD diets fed prepartum are acidogenic and aim to enhance the parathormone and vitamin D_3_ responses, increasing the blood Ca concentration at the beginning of the lactation [44]. Postpartum hypocalcemia is often referred to as a gateway condition that plays an essential role in the transition period, with later consequences for the dairy herd’s health and performance. However, subclinical hypocalcemia is not directly associated with pre- or postpartum metabolic stress. On the other hand, a recent publication by Wisnieski et al. [45] suggests that higher serum vitamin D_3_ concentrations at the dry-off and close-up periods were associated with increased urine ketone concentrations in early lactation. The authors mention that increased urine ketone concentrations may not be necessarily harmful, but it does indicate a greater degree of metabolic stress. They conclude that more studies should be conducted to establish definitive conclusions.

It is essential to mention that adaptation to different (prepartum) rations may induce changes in the length (and surface area) of ruminal papillae [46]. This has effects on the absorption of nutrients such as volatile fatty acids [46]. Moreover, it has been demonstrated that moving from a prepartum diet high in fiber and low in energy to a postpartum diet low in fiber and high in energy led to a shift in the ruminal microbiome [47]. These differences include a change in the relative abundance of cellulolytic and amylolytic bacteria. Significant shifts in the ruminal papillae, microbiome, and feed intake pattern associated with sudden diet changes may result in (mild) prepartum sub-acute ruminal acidosis, which may disrupt the ruminal epithelium, allowing the absorption of bacterial-derived lipopolysaccharides through the rumen [48]. In an attempt to neutralize free lipopolysaccharides in the bloodstream, the liver produces lipopolysaccharide binding protein (LBP), which is also recognized as an acute-phase protein [49]. Hence, high LBP levels in the prepartum period may contribute to systemic and/or dysregulation of inflammation. By dysregulation, we refer to an inflammatory response whose degree and/or duration impairs health rather than contributing to healing [50,51]. To the best of our knowledge, no studies have evaluated the effect of the (sudden) change from a far-off to a close-up diet on the gastrointestinal microbiome or the systemic inflammatory profile. The local and systemic effects of changing diets in the transition period (far-off to close-up) and its effects on future health are research topics that deserve further investigation.

### 3.2. Controlling the Length of the Dry Period

Multiple clinical trials have shown that shortening or omitting the dry period improved the energy [52] and metabolic status [24,53,54] of dairy cows in the subsequent lactation. Omitting the dry period increased the plasma concentrations of glucose, insulin, and insulin-like growth factor-1 and decreased the concentrations of plasma NEFA, BHB, and liver triacyl-glycerides [20,54], compared with those in cows with a traditional dry period length. Similarly, shortening the dry period (∼30 days) induced improvements in the postpartum metabolic status by decreasing plasmatic concentrations of NEFA and BHB compared to those in cows with an 8-week dry period [55]. Importantly, although omitting or shortening the dry period could have benefits in terms of energy balance, it often has adverse effects on udder health and milk yield [52,56], and increases the BCS of cows in the subsequent lactation [54]. Mayasari et al. [56] focused on the effect of dry period length on systemic inflammation and liver functionality. Omitting the dry period resulted in increased plasma concentrations of prepartum cholesterol and postpartum ceruloplasmin when compared with those observed with shortened (30 days) or traditional length (60 days) dry periods. The postpartum plasma concentrations of paraoxonase (a negative acute-phase protein) were also lower in cows with an omitted dry period in comparison to those in the other groups. However, the results of Mayasari et al. [56] could have been masked by the higher number of cows with postpartum clinical disease in the 0-day dry period group.

In a recent study, Ma et al. [53] tested the effect of a 0-day dry period length in combination with reduced postpartum dietary energy levels from 22 days postpartum onwards. Reducing the postpartum dietary energy level to match the lower milk yield following a 0-day dry period improved fertility in the cows with parity ≥3. This was accompanied by a better energy balance during the first seven weeks of lactation in comparison to that in cows with a traditional dry period length. This study suggests that the dry period length and the transition period diet could be tailored for cows in distinct lactations and with different degrees of body fattening at around 60 days before expected calving. This approach opens up new possibilities for managerial strategies at the dry-off to prevent over-conditioning and improve metabolic health in the early postpartum period. However, this research field is in a juvenile stage, and large-scale clinical trials are necessary to draw final conclusions.

## 4. Spontaneous Reduction in Feed Intake before Calving: The Chicken vs. the Egg Question

In one of the first papers concerning aspects of the dairy cow transition period, Drackley et al. [2] mentioned, “Our recent finding that prepartum intake has a major effect on postpartum DMI and periparturient lipid metabolism have raised many new questions about control of DMI, incidence of health disorders, and practical transition management”. Twenty years later, we are still struggling with the origin of the decreased DMI in the prepartum and its link with postpartum health. All dairy cows present a gradual decrease in feed intake associated with the growing physical space that the fetus occupies in the abdominal cavity as gestation evolves. However, around half of dairy cows experience a different manifestation of a sudden, transient decrease in feed intake in the last 2 weeks before calving, which has been associated with systemic inflammation, indicated by increased blood haptoglobin concentrations [57]. Despite its often-negative effects, the reasons for this reduction are still poorly understood. This prepartum systemic inflammation has been associated with a higher concentration of pro-inflammatory cytokines, including interleukin (IL)-1β, IL-6, and tumor necrosis factor (TNF)-α [58,59]. Johnson et al. [60] proposed that the release of pro-inflammatory cytokines has a role in worsening feed intake. However, the origin of these prepartum pro-inflammatory cytokines is still controversial, and the physiological rationale for the inflammation in the 1 to 2 weeks before calving escapes our understanding. Additionally, it is largely discussed whether cows stop eating because they suffer from systemic inflammation or whether the reduction in the feed intake is responsible for the decreased energy balance and concomitant systemic inflammation in the prepartum period (Figure 2).

Pro-inflammatory cytokines such as IL-1β, IL-6, and TNF-α are mainly released by macrophages [61], and these macrophages can be allocated to different organs such as the liver, brain, and adipose tissue [62]. The current hypothesis is that the fat mobilization associated with the reduction in DMI in the prepartum period is responsible for the release of these pro-inflammatory cytokines into the bloodstream. In the liver, these cytokines stimulate the production of positive acute-phase proteins (e.g., haptoglobin) and reduce the synthesis of negative acute-phase proteins (e.g., albumin) [63,64,65]. Trevisi et al. [59] proposed that a greater inflammatory status (based on IL-1β) starting already before calving is reflected in a lower liver functionality index in early lactation, based on changes in the concentrations of albumin, cholesterol, and bilirubin. High IL-1β concentrations are associated with a low liver functionality index in the last month of the dry period, which, in turn, seems to be related to the frequency of clinical diseases in the subsequent early lactation. Therefore, the understanding of the prepartum reduction in DMI is essential for improving the health and welfare of modern dairy cows.

In an attempt to gain further insight into the “chicken vs. egg” origin of inflammation and reduced feed intake in the prepartum period, Pascottini et al. [4] proposed a 4-day feed restriction model to mimic what is generally observed during the last 2 weeks before calving. The hypothesis was that feed restriction in the prepartum period would trigger fat mobilization and a subsequent release of proinflammatory cytokines and systemic inflammation. The prepartum 40% feed restriction model starting 2 weeks before expected calving produced significant fat mobilization as indicated by >2-fold greater NEFA concentration from the second until the last day of restriction (in comparison to that in the non-feed restricted herd mates). However, fat mobilization failed to generate systemic inflammation (based on haptoglobin and albumin concentrations). The authors therefore concluded that their feed restriction model failed to trigger systemic inflammation because it did not exactly replicate the pattern and duration of naturally occurring decreases in DMI in the prepartum period (at least in some cows). Therefore, the origin of the reduced feed intake in the prepartum period remained undetermined.

According to what has been discussed above, lipomobilization by itself does not trigger systemic inflammation in the prepartum period. However, feed restriction cannot be ruled out as a potential model to induce systemic inflammation. More extended or intensive feed restriction models could generate distinct results. On the other hand, feed restriction in over-conditioned cows may result in a different degree of fat mobilization that might be effective for the development of systemic inflammation. Nevertheless, a tentative conclusion could be that the naturally occurring reduction in DMI and the concomitant lipomobilization in the prepartum period cannot be regarded as the principal underlying cause of systemic inflammation in transition dairy cows.

## 5. Body Condition at the End of Pregnancy: The Turnover Point

Dairy cow over-conditioning at the end of pregnancy is associated with a plethora of metabolic, digestive, infectious, and reproductive conditions known as the “fat cow syndrome” [66,67]. The results and discussion presented in the present section represent a summary of studies performed by our research group [68,69,70,71,72,73]. For these studies, ten clinically healthy pregnant multiparous Holstein-Friesian cows were selected at the beginning of the dry period based on their BCS and classified as normal-conditioned (2.5–3.5; *n* = 5) or over-conditioned (3.75–5; *n* = 5). The phenotypes of these cows were clearly divergent, and over-conditioned cows had significantly higher backfat thickness (cm), serum NEFA (mmol/L), and adipose tissue weight (kg) than the normal-conditioned cows. The cows underwent multiple tests including the hyper-insulinemic euglycemic clamp test (HEC test) and the intravenous glucose tolerance test (IVGTT) at approximately three weeks before expected calving and were euthanized soon thereafter. The latter allowed us to eventually confirm the results found in vivo by in vitro tests on the harvested tissues following euthanasia. Detailed methodological data associated with management conditions, diets, and cow-related information are presented in the publications derived from these experimental cows [68,69,70,71,72,73]. Here, we synthesize the most relevant data found in terms of normal- and over-conditioning at the end of pregnancy on insulin resistance, adipose tissue physiology, and the liver transcriptomics profile.

### 5.1. Insulin Resistance at the End of Pregnancy

Insulin resistance is referred to as a state where a normal, physiological concentration of insulin results in a reduced biological response of insulin-sensitive tissues [74]. Insulin resistance can furthermore be subdivided based on two distinct features: insulin sensitivity and insulin responsiveness. The maximal effect of insulin determines the insulin responsiveness, while the concentration of insulin needed to elicit the half-maximal response determines the insulin sensitivity. Insulin resistance can hence be attributed to a decrease in insulin responsiveness (a downward shift of the insulin dose–response curve), a decrease in insulin sensitivity (a rightward shift of the insulin dose–response curve), or a combination of both [74]. At the end of pregnancy and in early lactation, all dairy cows experience a transient state of a decreased response to insulin in the peripheral tissues [74]. This homeorhetic adaptation serves as a mechanism to preserve a sufficient glucose supply for the fast-growing fetus and the mammary gland to ensure milk production [74,75]. These tissues drain most of the available glucose at this stage. However, reduced insulin sensitivity around calving could act as a double-edged sword. The exacerbated derailment of insulin’s action in peripheral tissues is known to contribute to pathological disorders such as subclinical and clinical ketosis, and fatty liver disease (and their consequences) [71]. Peripheral tissue function in the transition period relies on products of lipid metabolism such as NEFA and BHB [76]. However, to maintain homeostasis in the transition period, there should be an equilibrium between lipolysis and lipogenesis. In this regard, over-conditioned cows were believed to have an increased sensitivity toward lipolytic signals and a decreased sensitivity toward antilipolytic signals [77,78,79].

De Koster at al. [68] determined the insulin response of the glucose and fatty acid metabolism in over- and normal-conditioned cows at the end of pregnancy. In a novel set-up for dairy cows, cows underwent a HEC test three weeks before expected parturition. This test consisted of four consecutive insulin infusions with increasing insulin doses—0.1, 0.5, 2, and 5 mIU/kg per min. For each insulin infusion period, the steady state was defined as a period of 30 min where no (or minor) changes to the glucose infusion were necessary to keep the blood glucose concentration at the basal level. At each steady state, the glucose infusion rate and NEFA concentration were determined. With this model, De Koster at al. [68] showed a negative association between the level of fat accumulation (over-conditioning) and the insulin response of the glucose metabolism in pregnant dairy cows at the end of the dry period. This impaired insulin action at the level of the glucose metabolism was characterized by a decreased insulin sensitivity as well as a decreased insulin responsiveness, both being exacerbated in over-conditioned cows at the end of pregnancy. Conversely, the insulin response of the fatty acid metabolism was not associated with the level of fat accumulation in the dairy cows at the end of pregnancy. In other words, this means that even in over-conditioned cows, at the end of pregnancy, the lipolytic activity of insulin in the adipose tissue [80] is still active. Furthermore, it became clear that the response of fatty acid metabolism occurs at lower insulin concentrations than the response of glucose metabolism. The latter is logical and implies that in the fat tissue at the end of pregnancy, first of all, lipogenesis is downregulated, while lipolytic signals from insulin are still active. Hence, the insulin resistance of the glucose metabolism (downregulation of lipogenesis), in combination with the higher basal lipolysis, tips the delicate lipogenesis–lipolysis balance in favor of lipolysis, which causes lipomobilization to start before calving, especially in over-conditioned cows. However, since we could not demonstrate insulin resistance at the fat tissue level, we could not confirm the often-stated hypothesis that the massive lipomobilization, as encountered in many over-conditioned periparturient dairy cows, originates from a vicious circle based on insulin resistance in the obese fat tissue.

In a later study, Bogaert et al. [73] hypothesized that insulin resistance at the end of the dry period might be exacerbated by the pancreas’ insufficient secretory capacity due to pancreatic fat accumulation in over-conditioned cows. To prove this hypothesis, an IVGTT was first performed 3 weeks before expected calving, and then, cows were euthanized in the following week. After euthanasia, the pancreas was dissected and weighed, and pancreatic tissue samples were taken for histological analysis. The results revealed that over-conditioned cows presented greater fat infiltration in the pancreas than normal-conditioned cows. However, the pancreases of the over-conditioned were not heavier than those of the normal-conditioned cows. Interestingly, pancreatic islets from the over-conditioned cows presented an increase in size relative to the total area of pancreatic tissue. The greater functional tissue relative abundance of the pancreas in over-conditioned cows was reflected in the IVGTT. Over-conditioned cows presented greater insulin secretory capacity, as shown by the greater peak of insulin, greater acute insulin response to glucose, and greater area under the curve for insulin in comparison to such for normal-conditioned cows. A higher area under the curve for glucose during the first 60 min following the administration of the glucose bolus during the IVGTT confirmed the greater insulin resistance of the glucose metabolism of the peripheral tissues in the over-conditioned animals, despite the higher insulin secretion. These results suggest that the pancreas of over-conditioned cows compensates for the elevated level of peripheral insulin resistance by greater secretion of insulin.

Based on these studies, we concluded that insulin resistance in over-conditioned cows at the end of pregnancy is associated with impaired insulin action on glucose metabolism. By contrast, the NEFA-lowering effect of insulin was maintained, as evidenced by both in vivo and in vitro studies. This indicates that insulin resistance at the fat tissue level might not entail major complications in over-conditioned cows before calving. Furthermore, over-conditioned cows suffer from fat infiltration in the pancreas, although the latter was associated with greater pancreatic islets leading to a higher insulin secretory capacity as compensation for the concomitant peripheral insulin resistance.

### 5.2. Adipose Tissue Physiology at the End of Pregnancy

Reduced lipogenesis and enhanced lipolysis within adipocytes results in an elevated concentration of circulating NEFA around calving [81,82]. The release of NEFA at this time is essential for peripheral tissue metabolism, but excessive fat mobilization may result in elevated levels of ketone bodies and their health consequences in the postpartum period [83]. Over-conditioning (or the type of feeding as described by Drackley et al. [84]) at the end of pregnancy could lead to excessive lipomobilization due to the greater amount of fat available to mobilize in combination with an upregulated basal lipolytic activity [71]. It is clear that in over-conditioned cows, the absolute amount of deposited fat is greatly increased, and therefore, more fat is available to mobilize during lipolytic conditions. Nevertheless, De Koster et al. [71] challenged the hypothesis that basal lipolysis is upregulated in over- vs. normal-conditioned cows at the end of pregnancy. To do so, an in vitro explant culture of subcutaneous and omental adipose tissue was established from both groups of cows, and basal lipolysis was evaluated after the addition of insulin and catecholamines in different concentrations. The authors found that the adipocytes in the over-conditioned cows were enlarged, and these large adipocytes had an increased basal and catecholamine-stimulated lipolytic activity. Surprisingly, no clear evidence for a state of insulin resistance or the antilipolytic effect of insulin was found in large adipocytes, which was a clear confirmation of the results found in the in vivo study [68], in which exactly the same animals had been used. These results propose that the greater fat mobilization in over-conditioned cows is due to both a greater amount of available fat to mobilize and an upregulated basal lipolytic activity of the enlarged fat cells, especially in the subcutaneous depot. However, since the amount of fat accumulated in the abdomen is generally far greater than the subcutaneously deposited amount, the contribution of the abdominal fat to the overall fat mobilization phenomena is significantly higher. The latter was nicely evidenced by Hostens et al. [85], who found that in cows suffering from left displacement of the abomasum and concomitant negative energy balance, the profile of circulating fatty acids better resembled the profile of fatty acids accumulated in the omentum than that in the subcutaneous tissue.

Excessive fat deposition in combination with the concomitant increase in lipomobilization leads to the production of pro-inflammatory cytokines (mostly TNF-α and IL-6) [37,58]. This metabolic state (over-conditioning, lipomobilization, and pro-inflammatory cytokines) resembles sterile inflammation and metabolic syndrome in obese humans [86,87]. Interestingly, it was described that macrophages infiltrated within the adipose tissue are the main producers of the fat-derived pro-inflammatory cytokines in obese humans [88,89,90]. Depreester et al. [72] tested this hypothesis in a dairy cow model by determining the counts of adipose tissue macrophages (ATM) and the mRNA profile of TNF-α and IL-6 (among other adipokines) in subcutaneous and omental adipose depots from over- and normal-conditioned cows at the end of pregnancy. They found that over-conditioned cows had larger adipocytes, as demonstrated by the strong correlation between BCS and adipocyte size. Remarkably, the gene expression profiles of TNF-α, IL6, and leptin showed upregulation in adipose depots with enlarged adipocytes in the dairy cows at the end of pregnancy. Moreover, the number of ATM was greater in the omental adipose tissue with increased adipocyte size in comparison to that in the subcutaneous adipose tissue.

The primary function of ATM within the adipose tissue is to remove dead adipocytes, clear lipotoxic products, and stimulate adipogenesis [91]. However, ATM can express two phenotypes—M1, or classically activated ATM, and M2, or alternatively activated ATM. M1 ATMs are regarded as pro-inflammatory since they are responsible for the production of TNF-α and IL-6, while M2 ATMs possess anti-inflammatory properties (IL-10 secretion) [92]. Depending on the condition, ATM can be polarized to M1 or M2. Contreras et al. [93] demonstrated that the M1 polarization of ATM occurs in diseased cows (e.g., abomasal displacement with concurrent ketosis). Based on these findings, we hypothesize that over-conditioning may play a role in the M1 polarization of fat tissue ATM (mainly omental) and, thus, the upregulation of the pro-inflammatory (TNF-α and IL-6) transcripts. This could support the development of a low-grade inflammatory state, which may lead to a dysregulation of inflammatory processes in over-conditioned cows. However, more research is needed to confirm this hypothesis.

### 5.3. Global Gene Expression Profile of the Liver at the End of Pregnancy

In a last set of experiments, Pascottini et al. [94] tested if over-conditioning was associated with hepatic lipid accumulation and modifications in the hepatic global gene expression pattern at the end of pregnancy. Surprisingly, the total liver lipid percentage only tended to be greater in over- than in normal-conditioned cows. Even so, the liver of normal-conditioned cows had upregulated genes associated with enhanced functionality (albumin, selenoprotein P, and insulin-like growth factor binding protein 1 and 2), and over-conditioned cows had upregulated genes associated with inflammation (complement C3, hemopexin, and LBP). Interestingly, two outliers—one in normal-conditioned and another in over-conditioned cows, with high and low percentages of total liver lipids, respectively—were found. After removing these outliers, bioinformatics revealed that key genes associated with the acute phase response were upregulated in over-conditioned cows, including the genes encoding haptoglobin and LBP. The authors concluded that the level of transcription in the liver is associated with its lipid accumulation and that fatty liver mainly occurs in over-conditioned cows. However, high levels of lipid accumulation in the liver may also happen in normal-conditioned cows [95]. Individual cow variation regarding liver oxidative capacity plays an essential role in liver fat storage as a reflection of a cow’s adaptation to fat mobilization in the prepartum period. Moreover, using BCS as a proxy for over-conditioning might not always reflect the intra-abdominal adipose tissue mass [84]. Ji et al. [96] found that the visceral adipose tissue compared with subcutaneous adipose tissue seems to have a greater capacity for expression (and potentially secretion) of proinflammatory cytokines; thus, the excessive accumulation of visceral lipids may be detrimental to liver function and overall health, even in prepartum normal-conditioned cows.

### 5.4. Over-Conditioning and Spontaneous Reduction in Feed Intake

Based on the studies from our research group, it was concluded that in comparison to normal-conditioned cows, over-conditioned cows before calving have higher amounts of deposited fat containing greater adipocytes and greater numbers of infiltrated macrophages, with an upregulation of the TNF-α and IL-6 genes. Moreover, it was also demonstrated that the insulin resistance of the glucose metabolism, basal lipolysis, and NEFA concentrations were all greater in over- than in normal-conditioned cows. The latter was often associated with greater lipid accumulation in the livers of over-conditioned cows, with upregulation of the haptoglobin gene. Although we did not measure the feed intake of our experimental cows, we hypothesize that over-conditioning at the end of pregnancy (and all the above-described collateral effects) is associated with the spontaneous reduction in feed intake in the prepartum period in the 2 weeks before calving (Figure 3). Future research should be directed to demystify the role of the “fat cow syndrome” and the spontaneous reduction in feed intake in the prepartum period.

## 6. Conclusions

In comparison to other breeds (or species), in the last 20 to 30 years, the genetic selection of Holstein cows has mostly focused on prioritizing milk production over other physiological functions. We hypothesize that metabolic and inflammatory changes occurring from the dry-off until calving are of increased importance compared to those of dairy cows bred 30 years ago. The genetic selection in Holstein cows may have led to exacerbated metabolic changes in the adaptation to their increased milk production. Thus, it is of uttermost importance to have a metabolically healthy dairy cow at the end of pregnancy. However, it remains unknown whether the magnitude of metabolic and inflammatory changes in the dry period of modern dairy cows is a consequence of a physiological response or whether these changes have potential (deleterious) effects on their health and productive capacity. Our research group’s studies pointed out that over-conditioning at the end of pregnancy has significant effects on insulin resistance, fat tissue physiology, systemic inflammation, and liver functionality. The origin of the spontaneous reduction in feed intake 1 to 2 weeks before calving remains obscure. However, since BCS is associated with most of the metabolic and inflammatory alterations at the end of pregnancy, over-conditioning may be linked with the spontaneous reduction in feed intake and dysregulation of inflammation in the prepartum period.

## Figures and Tables

**Figure 1 animals-10-01419-f001:**
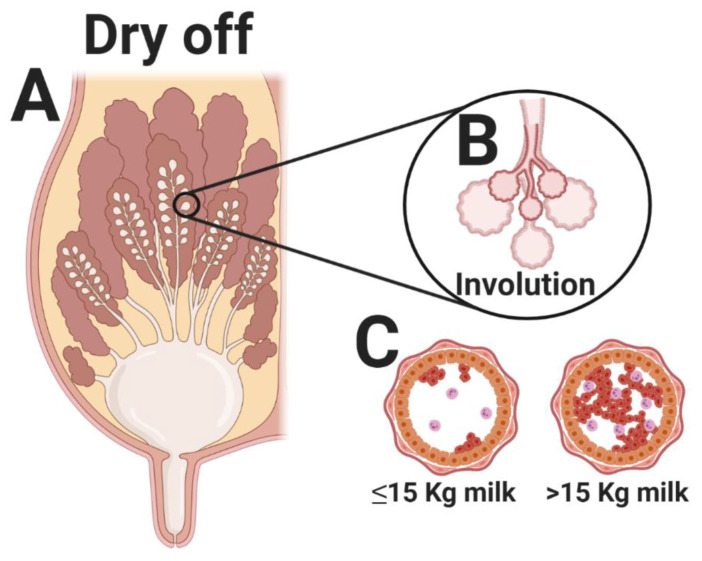
Schematic illustration of the local (and systemic) changes at the dry-off in dairy cows. (**A**) Milk cessation occurs abruptly 40 to 60 days before expected calving. (**B**) When milking is ceased abruptly, the cisternal ducts and alveoli of the udder become engorged, raising the intramammary pressure; these events induce the mammary gland involution process. (**C**) Mammary glands with lesser engorgement do exhibit an inflammatory response; however, the types and numbers of cells involved in the inflammatory response might be different when milk production at the dry-off is high. Mezzetti et al. [19] dichotomized the milk production at the dry-off as high (≥15 kg milk day) or low (<15 kg milk). Cows producing ≥15 kg of milk per day at the dry-off presented a greater inflammatory response than those producing <15 kg milk.

**Figure 2 animals-10-01419-f002:**
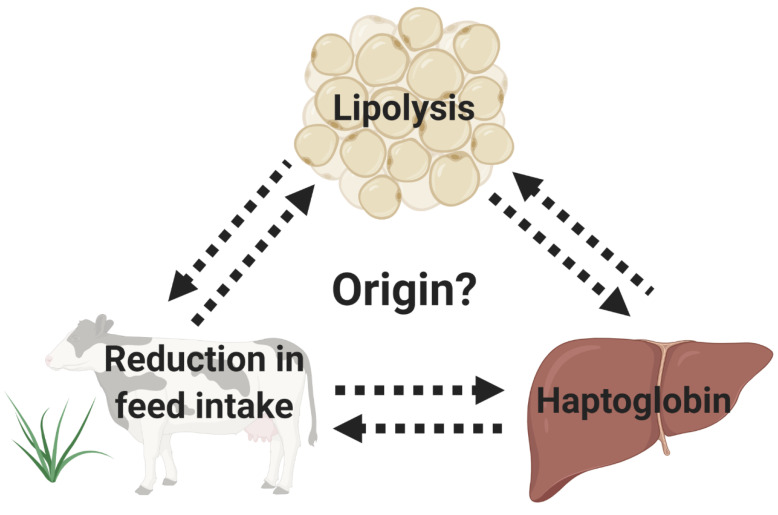
This schematic figure questions the origin of the systemic inflammation, lipolysis, and spontaneous reduction in feed intake in the prepartum period. It is largely discussed whether cows stop eating because they suffer from systemic inflammation or whether the reduction in the feed intake is responsible for the decreased energy balance and concomitant systemic inflammation in the prepartum period.

**Figure 3 animals-10-01419-f003:**
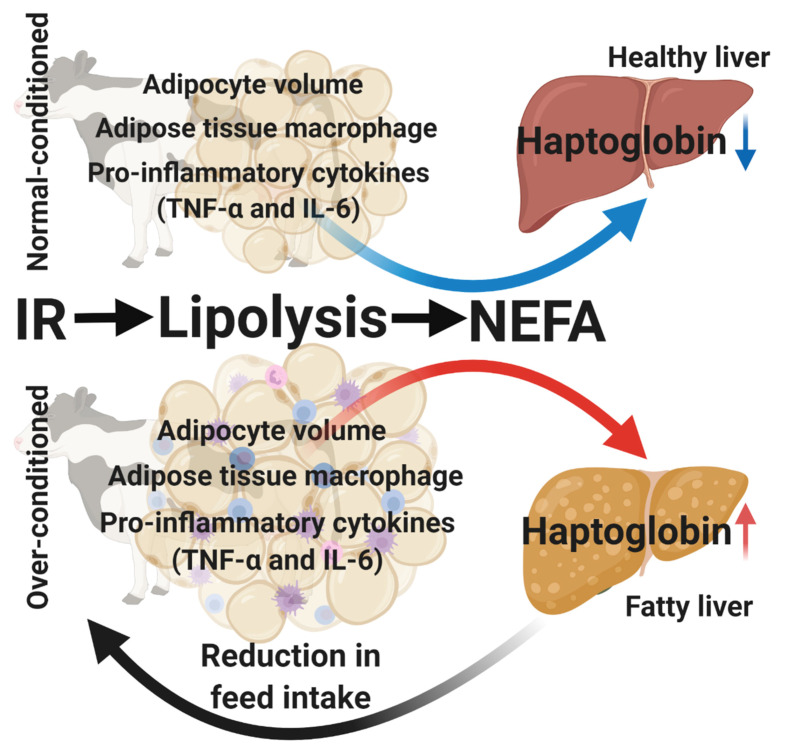
Schematic illustration of the potential origin of the spontaneous reduction in feed intake in the prepartum period in dairy cows. Based on the studies from our research group [68,69,70,71,72,73], it was demonstrated that over-conditioned cows before calving have a greater adipocyte volume, greater numbers of infiltrated macrophages, and upregulation of the tumor necrosis factor (TNF)-α and interleukin (IL)-6 genes in comparison to normal-conditioned cows. Moreover, it was also demonstrated that the insulin resistance (IR), basal lipolysis, and non-esterified fatty acid (NEFA) concentrations were greater in over- than in normal-conditioned cows. This was often associated with higher lipid accumulation in the livers of over-conditioned cows, with upregulation of the haptoglobin gene. Although we did not measure the feed intake of these cows, we hypothesize that over-conditioning cows is associated with the spontaneous reduction in feed intake in the prepartum period, triggered by their “fat cow syndrome’ in the 1 to 2 weeks before calving.

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
