# Peer review of "Metabolic Stress in the Transition Period of Dairy Cows: Focusing on the Prepartum Period"

_animals, 2020, doi:10.3390/ani10081419_

Round 1

Reviewer 1 Report

This review about prepartum metabolic (mal)adaptation in dairy cows provides a good overview about recent research and hypotheses in this field, with a focus on the authors’ research group. The manuscript is well written and has a clear structure. This reviewer has three questions and remarks and few minor corrections

1) This reviewer is missing information about genetic effects, i.e. are there any differences in metabolic adaptation between breeds or even in comparison with non-domestic ruminants? This would be interesting to know and could contribute to the general discussion about limits of milk yield, breeding goals and animal welfare.  

2) Are there any studies about the effect of the fetus on the maternal metabolism (related to this paper)?

3) L 197 and elsewhere: Could you speculate about the physiological reasons for prepartum “inflammation”? Is there a biological need for inflammation or inflammatory pathways before or at parturition. One example could be the switch from protecting the fetus and the fetal membranes to expelling the calf and the placenta?

Minor comments and correction:

L 30: Separate insulin resistance and transition disease with a semicolon.

L 35 Why “reprogramming”, not “programming”? Consider to replace with “adaptation”.

L 55: “gold standard”, not “golden standard”.

Fig 1: Remove “Engorgement Apoptosis Inflammation” from the picture. This is not self-explanatory and not well explained in the legend.

L 92-94: The second sentence refers to the first sentence, but different references are indicated.

L 97: This reviewer would assume that inflammatory response and pain are due to intramammary pressure on the day after drying-off rather than on mammary gland involution. Does involution start already at the first day?

L 143: Delete comma.

L 146: Does this sentence refer to SARA pre or post partum?

L 193: My understanding was that the decreased DMI is (among others) a result of the limited space in the abdomen due to the increasing size of the fetus?

L 221: Replace “eating” with “feed intake”

L 356: Delete comma.

L 414: Add reference (for quick readers who focus on the figures)

Author Response

Manuscript ID: animals-886630 " Metabolic stress in the transition period of dairy cows: focusing on the prepartum period"

We would like to thank the reviewers for their thorough reading and constructive remarks. We have modified the manuscript in response to the reviewers’ comments. Therefore, it is our hope that this revised manuscript will be acceptable for publication in Animals.

Reviewer #1

This review about prepartum metabolic (mal)adaptation in dairy cows provides a good overview about recent research and hypotheses in this field, with a focus on the authors’ research group. The manuscript is well written and has a clear structure. This reviewer has three questions and remarks and few minor corrections

1) This reviewer is missing information about genetic effects, i.e. are there any differences in metabolic adaptation between breeds or even in comparison with non-domestic ruminants? This would be interesting to know and could contribute to the general discussion about limits of milk yield, breeding goals and animal welfare. 

AU: The reviewer is correct. We should have been more specific in mentioning for which type of cow this review is focused. This review does not aim to compare genetic backgrounds, breed, nor species. Therefore, in key sections of the manuscript (simple summary, abstract, introduction, and dicussion) now we clearly mention “high-yielding dairy cows”, and in the introduction is specifically referred to as “(mostly) Holstein cows”. Furthermore, based on the comments of Reviewer 1, in lines 489 to 497 we introduced some philosophical discussion regarding genetic selection in Holstein cow and a (potential) increased importance of metabolic health in the prepartum period.

2) Are there any studies about the effect of the fetus on the maternal metabolism (related to this paper)?

AU: This is a remarkably interesting, but largely understudied question. To the best of our knowledge, there are not peer-reviewed publications on the topic. Our group is currently interested in studying the effect of fetal sex, genetic background, and breed (in Belgium, Holstein cows are often inseminated with Belgian Blue semen) on the metabolism and further productivity of the dam. However, currently, we are not able to answer the question raised by the reviewer. On the other hand, research by our group demonstrates that the metabolic state of the cow (and environmental factors) may affect the physiology of the placenta and the productivity of the future calf. We invite Reviewer 1 to check the following (recent) PhD thesis and review paper:

  • Van Eetvelde 2020. The impact of the prenatal environment on later performance in dairy cattle. https://biblio.ugent.be/publication/8656500
  • Opsomer et al. 2017. Epidemiological evidence for metabolic programming in dairy cattle. https://doi.org/10.1071/RD16410

3) L 197 and elsewhere: Could you speculate about the physiological reasons for prepartum “inflammation”? Is there a biological need for inflammation or inflammatory pathways before or at parturition. One example could be the switch from protecting the fetus and the fetal membranes to expelling the calf and the placenta?

AU: The physiological rationale of inflammation in the 1 to 2 weeks before calving escapes our understanding (now mentioned in lines 253 to 254), therefore we would prefer to do not raise potential erroneous hypothesis ins this aspect.

Minor comments and correction:

L 30: Separate insulin resistance and transition disease with a semicolon.

AU: Done (line 31).

L 35 Why “reprogramming”, not “programming”? Consider to replace with “adaptation”.

AU: Replace by adaptations (line 36).

L 55: “gold standard”, not “golden standard”.

AU: Done.

Fig 1: Remove “Engorgement Apoptosis Inflammation” from the picture. This is not self-explanatory and not well explained in the legend.

AU: Done.

L 92-94: The second sentence refers to the first sentence, but different references are indicated.

AU: The reference was corrected.

L 97: This reviewer would assume that inflammatory response and pain are due to intramammary pressure on the day after drying-off rather than on mammary gland involution. Does involution start already at the first day?

AU: This sentence was modified, “increased intramammary pressure” was added in line 99.

L 143: Delete comma.

AU: Deleted.

L 146: Does this sentence refer to SARA pre or post partum?

AU: Indeed, we hypothesize that a shift from a high to a low energy diet may in the prepartum may be associated with mild SARA. This is now mentioned in lines 197 to 198.

L 193: My understanding was that the decreased DMI is (among others) a result of the limited space in the abdomen due to the increasing size of the fetus?

AU: The reviewer is correct. The gradual decreased in feed intake associated with the increased size of the fetus at the end of pregnancy is now clearly distinguished from the sudden and transit decrease in feed intake associated with systemic inflammation (lines 243 to 246).

L 221: Replace “eating” with “feed intake”

AU: Done (line 278).

L 356: Delete comma.

AU: Deleted.

L 414: Add reference (for quick readers who focus on the figures)

AU: Done (line 478).

Reviewer 2 Report

This review explores an important and timely concept but before a final decision can be made several concerns have to be addressed.

As the authors are focusing on the pre-partum period, milk fever prevention using a negative DCAD approach is completely missing. Hypocalcemia around parturition is considered as one of the major metabolic diseases in transition dairy cows. It has been shown in 2 recent meta-analyses (Santos et al., 2019 and Lean et al., 2019) that feeding an acidogenic diet prepartum has beneficial effects on milk yield and transition cow health. The authors should include a chapter on milk fever prevention using a negative DCAD diet

In addition, the authors seem to ignore a large body of evidence that shows the positive impact of controlled energy diets on various aspects of metabolic health (authors are referred to…Drackley and Cardoso, 2014 Animal. 2014 May;8 Suppl 1:5-14; Cardoso et al., 2020 J Dairy Sci. 2020 Jun;103(6):5684-5693; Mann et al., 2015 J Dairy Sci. May;98(5):3366-82; Dann et al., 2006 J Dairy Sci. Sep;89(9):3563-77; Janovick et al., 2010 J. Dairy Sci. 94:1385–1400). It is questionable to this reviewer whether focusing ruminal adaptation is a major challenge when cows are fed controlled energy diets compared to the positive effects on energy metabolism. Besides prepartum energy level it has been shown that starch concentration and digestibility of the fresh cow ration have an impact on metabolism and inflammation (authors are referred to…Albornoz et al., 2018, 2019, 2020).

Specific Comments

L 60                The authors should also highlight impact of different dry-off procedure on animal welfare (authors are referred to… Zobel, G., D. M. Weary, K. E. Leslie, and M. A. G. von Keyserlingk. 2015. Invited review: Cessation of lactation: Effects on animal welfare. J. Dairy Sci. 98:8263–8277).

L 92                Coming from the same group of authors there was a recent publication indicating that Vitamin D levels at drying off are associated with postpartum BHB (authors are referred to… Wisnieski et al., 2020; J Dairy Sci Feb;103(2):1795-1806).

L 93                Lipolysis might also be a consequence of energy restriction when cows are switched to a dry cow diet after drying off.

L 127-156       See general comments. The authors should discuss the evidence for positive effects of controlled energy diets in the dry period on postpartum energy metabolism (i.e., NEFA, BHB, and liver TAG).

L174               Citation number 34 is a paper by Chen et al.

L 185              See general comments. The authors should discuss the evidence for positive effects of controlled energy diets in the dry period on dry matter intake around parturition. When feeding controlled energy diets it has been shown that dry matter content of the TMR (Havekes et al., 2020 J Dairy Sci. Feb;103(2):1500-1515) and chopping length of the straw (Havekes et al., 2020 J Dairy Sci. Jan;103(1):254-271) are important to support high DMI around parturition. Additionally, the results from a recent study (Havekes et al., 2020 J Dairy Sci. Jun;103(6):5070-5089) suggest that supplementing a molasses-based liquid feed in high-straw dry cow diets may improve intake and consistency in nutrients consumed during the dry period and in early lactation, as well as possibly promoting better rumen health across the transition period.

L 293              Please clarify. How does insulin act as a lipolytic signal?

L 331-355       Apart from BCS at drying off feeding strategy in the dry period may lead to higher abdominal fat accumulation (authors are referred to Drackley et al., 2014 J Dairy Sci.;97(6):3420-30).

L 383-398        The visceral adipose tissue compared with subcutaneous adipose tissue seem to have a greater capacity for expression (and potentially for secretion) of proinflammatory cytokines; thus, excessive accumulation of visceral lipid due to a long-term overfeeding energy may be detrimental to liver function and overall health of dairy cows, particularly during the transition period (Ji et al., 2014 J Dairy Sci.; 97(6):3441-8). Using BCS as a proxy for over-conditioning might not reflect intraabdominal mass of adipose tissue (Drackley et al., 2014 J Dairy Sci.;97(6):3420-30).

Author Response

Manuscript ID: animals-886630 " Metabolic stress in the transition period of dairy cows: focusing on the prepartum period"

We would like to thank the reviewers for their thorough reading and constructive remarks. We have modified the manuscript in response to the reviewers’ comments. Therefore, it is our hope that this revised manuscript will be acceptable for publication in Animals.

Reviewer #2

This review explores an important and timely concept but before a final decision can be made several concerns have to be addressed.

As the authors are focusing on the pre-partum period, milk fever prevention using a negative DCAD approach is completely missing. Hypocalcemia around parturition is considered as one of the major metabolic diseases in transition dairy cows. It has been shown in 2 recent meta-analyses (Santos et al., 2019 and Lean et al., 2019) that feeding an acidogenic diet prepartum has beneficial effects on milk yield and transition cow health. The authors should include a chapter on milk fever prevention using a negative DCAD diet

AU: We agree with the reviewer and a short paragraph regarding calcium homeostasis and negative DCAD diets has been added in lines 171 to 190. We acknowledge that this paragraph is superficial, therefore we explicitly mention that recent information more focused on the topic is available in recent publications (lines 171 to 173).

In addition, the authors seem to ignore a large body of evidence that shows the positive impact of controlled energy diets on various aspects of metabolic health (authors are referred to…Drackley and Cardoso, 2014 Animal. 2014 May;8 Suppl 1:5-14; Cardoso et al., 2020 J Dairy Sci. 2020 Jun;103(6):5684-5693; Mann et al., 2015 J Dairy Sci. May;98(5):3366-82; Dann et al., 2006 J Dairy Sci. Sep;89(9):3563-77; Janovick et al., 2010 J. Dairy Sci. 94:1385–1400). It is questionable to this reviewer whether focusing ruminal adaptation is a major challenge when cows are fed controlled energy diets compared to the positive effects on energy metabolism. Besides prepartum energy level it has been shown that starch concentration and digestibility of the fresh cow ration have an impact on metabolism and inflammation (authors are referred to…Albornoz et al., 2018, 2019, 2020).

AU: A paragraph with very recent literature regarding feeding controlled energy diets in the prepartum was added in lines 144 to 170 as suggested by the reviewer. We also mention in lines 129 to 130 that recent literature more focused in feeding strategies are available other recent publications.

Specific Comments

L 60 The authors should also highlight impact of different dry-off procedure on animal welfare (authors are referred to… Zobel, G., D. M. Weary, K. E. Leslie, and M. A. G. von Keyserlingk. 2015. Invited review: Cessation of lactation: Effects on animal welfare. J. Dairy Sci. 98:8263–8277).

AU: We added “including dairy cow welfare” and we cited the paper of Zobel et al. in line 59. For this review we only focus on prepartum metabolic stress and we would prefer not to include animal welfare issues.

L 92 Coming from the same group of authors there was a recent publication indicating that Vitamin D levels at drying off are associated with postpartum BHB (authors are referred to… Wisnieski et al., 2020; J Dairy Sci Feb;103(2):1795-1806).

AU: A brief mention of the paper of Winieski et al. was added in lines 185 to 190.

L 93 Lipolysis might also be a consequence of energy restriction when cows are switched to a dry cow diet after drying off.

AU: Correct. “energy restriction (cows switched to a dry cow diet after drying-off)” was added in line 96.

L 127-156 See general comments. The authors should discuss the evidence for positive effects of controlled energy diets in the dry period on postpartum energy metabolism (i.e., NEFA, BHB, and liver TAG).

AU: As mentioned before, a paragraph with very recent literature regarding feeding controlled energy diets in the prepartum was added in lines 144 to 170.

L174 Citation number 34 is a paper by Chen et al.

AU: Corrected.

L 185: See general comments. The authors should discuss the evidence for positive effects of controlled energy diets in the dry period on dry matter intake around parturition. When feeding controlled energy diets it has been shown that dry matter content of the TMR (Havekes et al., 2020 J Dairy Sci. Feb;103(2):1500-1515) and chopping length of the straw (Havekes et al., 2020 J Dairy Sci. Jan;103(1):254-271) are important to support high DMI around parturition. Additionally, the results from a recent study (Havekes et al., 2020 J Dairy Sci. Jun;103(6):5070-5089) suggest that supplementing a molasses-based liquid feed in high-straw dry cow diets may improve intake and consistency in nutrients consumed during the dry period and in early lactation, as well as possibly promoting better rumen health across the transition period.

AU: All the papers published from the Master Thesis of Casey Havekes et al. are now briefly discussed in lines 158 to 170.

L 293: Please clarify. How does insulin act as a lipolytic signal?

AU: At this stage, the content of this review is already too dense. We prefer to add a reference here to support the idea (line 352). In the study by De Koster et al. it is very nicely described the hormone sensitive lipase contribution to adipose tissue lipolysis and how insulin regulates its activity in periparturient dairy cows 11 d prepartum (dry) and 11 (fresh) and 24 d (lactation) postpartum.

L 331-355: Apart from BCS at drying off feeding strategy in the dry period may lead to higher abdominal fat accumulation (authors are referred to Drackley et al., 2014 J Dairy Sci.;97(6):3420-30).

AU: This reference is now briefly mentioned in line 393.

L 383-398: The visceral adipose tissue compared with subcutaneous adipose tissue seem to have a greater capacity for expression (and potentially for secretion) of proinflammatory cytokines; thus, excessive accumulation of visceral lipid due to a long-term overfeeding energy may be detrimental to liver function and overall health of dairy cows, particularly during the transition period (Ji et al., 2014 J Dairy Sci.; 97(6):3441-8). Using BCS as a proxy for over-conditioning might not reflect intraabdominal mass of adipose tissue (Drackley et al., 2014 J Dairy Sci.;97(6):3420-30).

AU: We would like to thank the reviewer for this rationale whi
